# Emerging Mechanisms of Pulmonary Vasoconstriction in SARS-CoV-2-Induced Acute Respiratory Distress Syndrome (ARDS) and Potential Therapeutic Targets

**DOI:** 10.3390/ijms21218081

**Published:** 2020-10-29

**Authors:** Harry Karmouty-Quintana, Rajarajan A. Thandavarayan, Steven P. Keller, Sandeep Sahay, Lavannya M. Pandit, Bindu Akkanti

**Affiliations:** 1Department of Biochemistry and Molecular Biology, McGovern Medical School at The University of Texas Health Science Center at Houston, Houston, TX 77030, USA; 2Divisions of Pulmonary, Critical Care and Sleep Medicine, McGovern Medical School at The University of Texas Health Science Center at Houston, Houston, TX 77030, USA; bindu.h.akkanti@uth.tmc.edu; 3Department of Cardiology, Houston Methodist Hospital, Houston, TX 77030, USA; ramirthalingamthandavarayan@houstonmethodist.org; 4Division of Pulmonary and Critical Care Medicine, Brigham and Women’s Hospital and Harvard Medical School, Boston, MA 02115, USA; SKELLER@BWH.HARVARD.EDU; 5Co-Director, Pulmonary Vascular Diseases Center, The Methodist Hospital, Houston, TX 77030, USA; ssahay@houstonmethodist.org; 6Michael E. DeBakey Veterans Affairs Medical Center, Baylor College of Medicine, Houston, TX 77030, USA; lpandit@bcm.edu

**Keywords:** acute lung injury, respiratory viral infection, COVID-19, renin angiotensin system, Kallikrein–Kinin System, cytokine release storm, hypoxic-adenosinergic response, endothelin

## Abstract

The 1918 influenza killed approximately 50 million people in a few short years, and now, the world is facing another pandemic. In December 2019, a novel coronavirus named severe acute respiratory syndrome coronavirus 2 (SARS-CoV-2) has caused an international outbreak of a respiratory illness termed coronavirus disease 2019 (COVID-19) and rapidly spread to cause the worst pandemic since 1918. Recent clinical reports highlight an atypical presentation of acute respiratory distress syndrome (ARDS) in COVID-19 patients characterized by severe hypoxemia, an imbalance of the renin–angiotensin system, an increase in thrombogenic processes, and a cytokine release storm. These processes not only exacerbate lung injury but can also promote pulmonary vascular remodeling and vasoconstriction, which are hallmarks of pulmonary hypertension (PH). PH is a complication of ARDS that has received little attention; thus, we hypothesize that PH in COVID-19-induced ARDS represents an important target for disease amelioration. The mechanisms that can promote PH following SARS-CoV-2 infection are described. In this review article, we outline emerging mechanisms of pulmonary vascular dysfunction and outline potential treatment options that have been clinically tested.

## 1. Introduction

Since December 2019, severe acute respiratory syndrome coronavirus 2 (SARS-CoV-2), a member of the beta-coronavirus family, has spread globally, leading to a pandemic of coronavirus disease 2019 (COVID-19) [1]. Over 42.5 million cases and 1.1 million deaths have been reported globally (https://coronavirus.jhu.edu/accessed 25 October 2020). Since the first reported case in the USA in 19 January 2020 [2], over 8.6 million cases and over 225,000 deaths have been reported in the USA (https://coronavirus.jhu.edu/accessed 25 October 2020). COVID-19 is likely to remain a dangerous threat to human health until an effective vaccine or therapy is developed.

Although the majority of COVID-19 cases are considered mild, a subset of patients develops severe respiratory symptoms with high viral load [1,2,3]. These patients typically develop dyspnea and hypoxemia and rapidly progress to acute respiratory distress syndrome (ARDS) and multi-organ failure [1]. A report from Wuhan, China demonstrated that 41.8% of patients admitted with COVID-19 developed ARDS, of which 33% required mechanical ventilation. The authors report that all patients who died from COVID-19 developed ARDS and received mechanical ventilation [4]. Similarly, a report from 5700 patients in the New York City Area reported an accelerated mortality rate of 88.1% in patients that developed ARDS and required mechanical ventilation [5]. These observations highlight the development of ARDS as a principal cause of death in patients with COVID-19. Importantly, there is a lack of effective therapies for the management of COVID-19 [6].

ARDS is the clinical presentation of acute lung injury; it is defined by the presence of bilateral opacities from chest imaging, non-cardiogenic pulmonary edema, acute hypoxemia, reduced lung compliance, and compromised arterial oxygenation calculated by the partial pressure of oxygen (Pa_O2_)/fraction of inspired oxygen (Fi_O2_) ratio. A ratio less than 300 mmHg is termed mild, less than 200 is termed moderate, and less than 100 is severe [7,8,9,10]. The severity of ARDS correlates directly with mortality, and currently, there are no effective pharmacological treatments for ARDS, as painfully highlighted by the COVID-19 pandemic. 

Importantly, patients with ARDS can develop pulmonary hypertension (PH), which is defined by increased pulmonary vascular resistance (PVR) ≥ 3 WU associated with an increased mean pulmonary arterial pressure (mPAP > 20 mmHg) at rest [11]. The increased mPAP contributes to right ventricle hypertrophy (RVH) and right-sided heart failure [12]. There have been reports of patients with COVID-19 that present with right ventricular impairment on echocardiography with prevalence as high as 33% [13]. However, we have yet to understand the pathophysiology of right ventricular dysfunction and if it correlates to elevated pulmonary arterial pressures, as no right heart catheterization data is available on a larger scale. 

Pulmonary embolism is the most commonly reported thrombotic complication of COVID-19 [14]. Multiple studies have revealed that the incidence of venous thromboembolism is as high as 27% [15,16,17]. Multiple autopsy series have revealed pulmonary thromboses in patients with COVID-19, and whether this is a result of an initial endothelial injury or an independent event is not fully understood [18,19].

The development of PH in ARDS is of particular importance in patients with COVID-19 where a history of cardiovascular disease has been recognized as one of the most important risk factors outside of increasing age [20,21]. It also important to mention that infection of SARS-CoV-2 has also been implicated in promoting myocardial dysfunction in infected patients [21,22]. Since ARDS is also known to induce PH [11], and SARS-CoV-2 is known to promote injury to the pulmonary vasculature [23], understanding the potential mechanisms that may lead to PH in COVID-19 is important to identify novel treatments for this deadly disease. This review will focus on the potential mechanisms and therapeutic targets that contribute to PH in COVID-19-induced ARDS focusing in pulmonary vasoconstriction.

## 2. COVID-19 in Patients with Pulmonary Hypertension (PH)

At present, very few data are available regarding the severity of SARS-CoV-2 infection in patients with a diagnosis of PH. Most of the data recapitulated are from patients with World Health Organization (WHO) Group 1 PH or pulmonary arterial hypertension (PAH), whereas no reports (as of 25 October 2020) from patients with WHO groups 2–5 have been made available. From these early studies, only 13 patients with a dual diagnosis of PAH and COVID-19 have been reported in the US [24]. Although PAH is a rare disease, this number is significantly lower than expected and is likely due to the pandemic having just begun at that time point and will likely change as the COVID-19 cases continue to escalate [25]. It is also reasonable to assume that patients with PH in the presence of left-heart disease (Group 2 PH) or in the presence of chronic lung diseases (Group 3 PH) are at greater risk of COVID-19 than patients with cardiovascular or chronic lung diseases alone, since PH enhances morbidity and mortality in these patients [26]. Patients with chronic thromboembolic pulmonary hypertension (CTEPH) may represent an important risk group for COVID-19 patients [26], given the emerging threat of thrombosis in the pathophysiology of COVID-19 [27,28]. However, studies on the morbidity and mortality in different WHO PH groups are not yet available. 

Patients with PAH that develop COVID-19 require advanced management, as many of the treatment strategies such as ventilation can severely aggravate the respiratory insufficiency [29]. As such, several groups have identified special management strategies for PAH patients [26,29]. These strategies can be successful as demonstrated in a patient with severe PAH and β-Thalassemia Major who recovered from COVID-19 pneumonia [30]

## 3. Pulmonary Vascular Sequela in SARS-CoV-2 Infection

A recent and comprehensive review of the pulmonary vascular changes that occur following SARS-CoV-1 and SARS-CoV-2 infection has been recently published by Potus et al. [23]. In this review, the authors document how infection by SARS-CoV-2 can lead to similar pathophysiological findings with PH. Notably increased pro-inflammatory cytokines, angiotensin II, evidence of DNA damage—poly (ADP-ribose), polymerase (PARP), and hypoxic pulmonary vasoconstriction (HPV)—are most similar between COVID-19 and PH patients. In addition, evidence of thrombosis (D-dimers, prothrombin), cardiac injury (increased troponin and natriuretic peptide and RV dilatation), and increased pulmonary vascular thickness are also shared between COVID-19 and PH, as is dyspnea [23].

In an autopsy series that has recently been published, the lungs of patients with COVID-19 when compared to those of H1N1 influenza complicated by ARDS had three distinctive angiocentric features of COVID-19 [31]. They report that severe endothelial injury associated with intracellular SARS-CoV-2 virus resulted in disrupted endothelial cell membranes, in addition to vascular thrombosis with microangiopathy and occlusion of the alveolar capillaries. The third most significant finding from this recent series was unexpected new vessel growth through the mechanism of intussusceptive angiogenesis [31]. 

Intussusceptive angiogenesis, in contrast to sprouting angiogenesis, has been recently described in greater detail in the last few decades. Initially described in the setting of new capillary network growth, the mechanism consists of repeated insertion of new slender transcapillary tissue pillars. Two endothelial leaflets come into close contact, form new junctional complexes, and then these thin out to give way to invading interstitial tissue such as fibroblasts, myofibroblasts, and pericytes. One could speculate if the initial ground glass lesions seen in early COVID-19 on high-resolution computed tomography images are early signs of intussusceptive angiogenesis and breakdown of the capillary membrane with subsequent interstitial edema [32]. This process has some similarity with endothelial dysfunction in PH that leads to progressive vascular remodeling [33,34] and may exacerbate to the loss of gas-exchange capacity of the lung in COVID-19.

## 4. Mechanisms of Vasoconstriction in COVID-19

### 4.1. The Renin–Angiotensin System (RAS)

A central pathway in the pathophysiology of COVID-19 is the renin–angiotensin system (RAS). The RAS is a hormonal system that regulates many pathophysiological events in the body including vascular pressure [35,36,37]. The RAS can be broadly divided into two major arms: the angiotensin converting enzyme (ACE), angiotensin II (Ang II), and the angiotensin type 1 (AT_1_) receptor; and the angiotensin converting enzyme 2 (ACE2) and angiotensin-(1–7) [Ang-(1–7)] [37,38]. Importantly for SARS-CoV-2 infection, the ACE2 enzyme serves as the functional receptor for the SARS-CoV-2 to enter the cell through interaction with the transmembrane protease serine 2 (TMPRSS-2) [39]. Infection typically starts in the epithelia of the upper respiratory tract before spreading to the alveoli [40,41], where alveolar type II cells are known to express high levels of ACE2 [42]. Following SARS-CoV-2 infection, it is widely accepted that the formation of the ACE2/virus complex results in reduced ACE2 activity and expression [43,44]. This functional loss of ACE2 activity leads to a dysregulation in the RAS, resulting in reduced generation of Ang-(1–7) and the subsequent accumulation of Ang II [45]. This is an important pathological step in COVID-19 since Ang-(1–7) is associated with beneficial effects such as vasodilation and it suppresses inflammation; however, Ang II has potent hostile effects by regulating inflammation and injury [46]. Importantly, the activation of angiotensin II receptor type 1 receptor (AT_1_) receptor by Ang II is known to mediate potent vasoconstrictor responses [38]. ACE2 has been found to be highly expressed in multiple tissues in addition to respiratory epithelial cells, including myocardial cells, and the proximal tubule cells of the kidney and the gut epithelium [47], which may explain the clinical presentation of COVID-19. Several recent reviews have summarized the effect of SARS-CoV-2 on the RAS, resulting in an imbalance of ACE1/ACE2 [45,48,49,50,51]. Importantly, this process has also been documented in the vascular endothelium, including the pulmonary vasculature [52]. Thus, we hypothesize that this ACE1/ACE2 imbalance results in the accumulation of Ang II in the pulmonary vasculature that can promote pulmonary vasoconstriction responses. This phenomenon is consistent with known roles of the RAS in PH, where decreased ACE2 is observed in patients with PH [53]. An overexpression of ACE2 by recombinant ACE2 or small molecules were reported to attenuate PH by regulating angiotensin-(1–7) signaling and NO bioavailability [54]. The role of the RAS has also been shown in models of PH in the presence of lung fibrosis where recombinant human (rh)ACE2 reduced bleomycin-induced PH [55]. Although Ang II inhibitors are not used to treat PH, endothelin receptor antagonists (ERAs) are known to reduce Ang II levels [56] and have been shown to be beneficial in viral-induced inflammation [57]. 

### 4.2. The Kallikrein–Kinin System (KKS)

An important downstream effect resulting from the imbalance of the RAS is the subsequent alterations on the Kallikrein–Kinin System (KKS). The KKS is a complex hormonal system that regulates many processes including blood pressure, inflammation, fibrosis, and pain [58]. These effects are mediated by cleavage of plasma (high molecular weight, HMW) or tissue (low molecular weight) kallikrein by kininogen to generate different kinins, which are oligopeptides containing the amino acid sequence of bradykinin. Importantly, the cleavage of HMWK (plasma) by kininogens yields bradykinin and the cleavage of LMWK (tissue) yields lys-bradykinin, which is also known as kallidin [58,59]. Bradykinin or kallidin activate bradykinin (B)_1_ or B_2_ receptors to mediated their effects [58]. 

Following imbalance of the ACE1/ACE2 system due to SARS-CoV-2 infection, the depletion of ACE2 leads to an accumulation of a bioactive peptide [des-Arg^9^]bradykinin, which is normally cleaved by ACE2 [59,60]. [des-Arg^9^]bradykinin is known to modulate lung injury and inflammation [61,62] through the activation of both B_1_ and B_2_ receptors [60]. The effects of [des-Arg^9^]bradykinin have been shown to be blocked by icatibant, which is a selective antagonist for the B_2_ receptor [63,64] that is FDA approved to treat angioedema. B_2_ receptor blockade has also been shown to inhibit smooth muscle contraction in models of experimental airway hyperresponsiveness [63,64], and both B_1_ and B_2_ receptors are known to induce vasoconstriction [65,66]. 

In addition to the effects of [des-Arg^9^]bradykinin due the depletion of ACE2, an increased accumulation of bradykinin may be probable in COVID-19 [67,68]. Herein, accumulating Ang II levels due to a loss of ACE2 can result in the reduced enzymatic activity of ACE [68] that cleaves Ang II from Ang I [58]. Bradykinin derived from plasma kallikrein is usually broken down by ACE1 [59]; thus, a reduced activity of ACE1 will result in an accumulation of bradykinin in addition to [des-Arg^9^]bradykinin. Although bradykinin is known to modulate vasodilation systemically [58,67], it has also been shown to modulate pulmonary vasoconstriction in experimental models of acute lung injury [69,70]. Thus, it is plausible that in COVID-19, elevated bradykinin and [des-Arg^9^]bradykinin not only promote inflammation and cough but also modulate important pulmonary vasoconstriction effects in severe patients contributing to PH. In support of this, B_2_ blockade have been shown to experimentally reduce experimental PH in rats [71]. Thus, we hypothesize that drugs such as icatibant that block the B_2_ receptor or lanadelumab—a monoclonal antibody against HMWK that yields bradykinin, that is also FDA approved for the treatment of angioedema—may represent important avenues that may not only inhibit the inflammatory cascade of the KKS [59,67] but may also inhibit the vasoconstrictor effect of bradykinin. 

## 5. Drugs Targeting the RAS

ACE inhibitors (ACEi) and angiotensin receptor blockers (ARBs) are commonly prescribed to manage systemic hypertension, and thus, controversy exists regarding the impact of these drugs in patients that get infected with SARS-CoV-2 [72]. Studies in COVID-19 have reported lower mortality rates in patients receiving therapy with ACEi and ARBs [73], prompting physicians to advise in favor of maintaining ACEi and ARB therapy in patients with COVID-19 [74]. ACEi and ARBs work to lower blood pressure by inhibiting the synthesis and the effects of AngII [45]. In addition, ACEi and ARBs have also been shown to be beneficial in experimental models of acute lung injury [45]. The ACEi captopril was able to inhibit experimental lung injury induced by oleic acid and endotoxin challenge [75,76]. The ARB losartan also inhibited inflammation in endotoxin-treated mice [77]. These studies highlight the important role of AngII in acute lung injury and are in line with evidence of AngII upregulation playing a central role in coronavirus-mediated ARDS [44,51,78]. Two clinical trials testing Losartan in patients in COVID-19 are underway (NCT0431117 and NCT04312009), highlighting the therapeutic potential of drugs targeting RAS. However, ACEi and ARBs are known to stimulate the upregulation of ACE2, which is the functional receptor that allows SARS-CoV-2 to infect cells [39]. Thus, treatment with ACE1 or ARBs may result in enhanced viral docking options for SARS-CoV-2. An important side effect of ACEi is the development of a dry cough resulting from an accumulation of bradykinin [79]. Dry cough is a common feature of COVID-19 [1]; thus, a potential complication of ACEi in COVID-19 may be the upregulation of bradykinin. Taken together, the potential beneficial effects of ACEi/ARB therapy in COVID-19 may be outweighed by either increased ACE2 expression or an accumulation of bradykinin. An alternative approach is the use of specific agonist Ang-(1–7) receptors, as previously addressed by a comprehensive review on the RAS in COVID-19 [80].

The central mechanisms that lead to PH following dysregulation of the RAS and potential therapeutic approaches are summarized in Figure 1.

## 6. Hypoxic Pulmonary Vasoconstriction

Hypoxic Pulmonary Vasoconstriction (HPV) is a unique feature of lung to ensure maximal gas exchange and blood oxygenation in the circulation [81]. Elsewhere in the body, the presence of hypoxia in the systemic circulation leads to vasodilation. Further, hypoxic conditions in the pulmonary circulation lead to vasoconstriction [82,83]. However, long-term HPV can lead to pulmonary hypertension (PH), which has systemic complications and increases the workload of the right ventricle, potentially leading to right ventricle failure. HPV is a reflex contraction of vascular smooth muscle, which can be triggered by several factors, and chief among them is alveolar P_O2_ [82,83]. A major site for the HPV response is in the pulmonary arteries, where smooth muscle cells sense and constrict in the setting of low levels of oxygen, resulting in an increase in pulmonary vascular resistance (PVR) and subsequent PH [84]. Recent murine studies also demonstrate that sustained hypoxic vasoconstriction is present only in the small intra-pulmonary arteries (80–200 µm in diameter) [85]. This finding has important implications, as it is the distal areas of the lung where changes in P_O2_ are most closely linked with gas exchange and where the shunting of blood flow is crucial.

Studies in healthy volunteers demonstrate that HPV occurs in two distinct phases: an initial phase responding within a few seconds, and a secondary phase in the setting of sustained hypoxia. The initial response occurs within seconds of moderate hypoxia (P_O2_ 30–50 mmHg), leading to a maximal PVR at 15 min. If hypoxic conditions persist for more than 30–60 min, a secondary increase in PVR is observed, reaching a peak at 2 h and lasting for at least 8 h [86,87]. This study also reported that even with a reinstitution of normoxic conditions, PVR does not immediately reverse to baseline levels, suggesting a complex mechanism that takes several hours to recover [86].

HPV is a unique response of the lung that has an important physiological role: it acts to match perfusion to ventilation to optimize P_O2_ [88]. Thus, shunting blood toward areas with high ventilation maintains efficient gas exchange processes. However, in situations where alveolar hypoxia is widespread, as in the case of high altitude or in diffuse lung disease, PH can develop [89]. In the setting of ARDS, PH is recognized as an important consequence of acute lung injury that can worsen prognosis [90,91]. In addition, RV failure can be encountered in 20–50% of ARDS patients, which is a critical cause of survival and is thought to be related to HPV and positive pressure ventilation, among other mechanisms [92]. Mechanistically, HPV is a highly complex process where changes in the redox state, increased generation of reactive oxygen species, and changes in the energy state of cells can lead to an inhibition of K^+^ channels and Ca^2+^ cellular influx, leading to vasoconstriction [93].

Recent clinical observations highlight an atypical presentation of ARDS in COVID-19 patients [94]. COVID-related ARDS presents rapidly and is accompanied by severe hypoxemia, requiring support by mechanical ventilation as described in multicenter case series reports of patients in the Seattle region [94]. In another case report, a patient with COVID-19 had sudden shortness of breath and hypoxia with pulmonary embolism by RV failure [95]. These observations suggest that in COVID-related ARDS, severe hypoxemia may contribute to rapid progression of disease. Based on these observations and the unique pathophysiology of the virus, it is possible for dysregulated HPV to play a central role in the pathogenesis and severity of ARDS in COVID-19. However, controversy exists regarding the exact role HPV in COVID-19 [23,83], and it is possible for this response to be amplified, leading to regional pulmonary vasoconstriction, contributing to PH, or for it to be dysregulated, resulting in an inability to adequately respond to perfusion–ventilation mismatch. 

## 7. High-Altitude Induced Pulmonary Edema 

A further complication of prolonged HPV highly relevant to ARDS is the development of high-altitude induced pulmonary edema (HAPE), which usually presents between 2 and 5 days following acute exposure to altitudes above 2500–3000 m. HAPE is characterized by patchy peripheral distribution of edema [96]. Based on this observation, it is conceivable that cases of severe ARDS, such as that observed in COVID-19, may promote HPV acting to trigger further edema formation similar to HAPE, thus producing a vicious spiral of worsening of disease. Indeed, earlier studies pointed at a link between the HAPE and COVID-19 cases [97]. However, it is important to note recent studies highlighting the differences in the etiology of HAPE and COVID-19 pneumonia; whereas HAPE is a result of excessive HPV leading to increased pulmonary capillary hydrostatic pressure and leakage, COVID-19-induced pneumonia is a result of viral-induced inflammation culminating in epithelial cell dysfunction and an accumulation of fluid in the alveolar spaces [83,98,99]. These important differences in the pathophysiology of HAPE vs. COVID-19-induced edema and the atypical presentation of severe hypoxemia but relatively well-maintained lung mechanics with severe capillary leak [100] add to the controversy of the mechanisms that lead to edema and pulmonary vasoconstriction in COVID-19 [83].

## 8. Nitric Oxide

Several mechanisms of HPV have been detailed in recent reviews [84,101]. Most relevant to SARS-Cov2-induced ARDS may be the nitric oxide (NO) pathway, as NO inhalation therapy has been employed in the management of patients with acute RV failure secondary to PH and ARDS [102,103]. Indeed, NO inhalation is currently being examined as a protective measure and treatment against COVID-19 (clinical trials: NCT04306393, NCT04312243, NCT04338828, NCT04305457). In patients with SARS-CoV, inhaled NO reversed PH and reduced the length of ventilator support. Additionally, in vitro observation has reported that coronaviruses are mostly vulnerable to NO, suggesting that NO may prevent the viral replication in coronavirus-associated SARS and decrease the lung cytokine storm [104,105]. Mechanistically, NO is a potent vasodilator that stimulates soluble guanylate cyclase (sGC) in vascular smooth muscle cells, increasing cyclic guanosine monophosphate (cGMP) to induce vasodilation [106]. The actions of cGMP are abrogated by phosphodiesterase 5 (PDE5), which is the most abundant PDE in the lung circulation [107]. In the context of HPV, inhaled NO is able to attenuate both the acute and sustained phase of HPV [108,109]. In another preclinical observation, inhaled NO reversed HPV and prevents the worsening of PH [110]. Additional studies demonstrate that PDE5 inhibition was able to attenuate the prolonged HPV in high-altitude hypoxia for 14 days [111] and prevent HAPE [112]; thus, in addition to NO, therapeutic PDE5 inhibition may also provide a benefit to patients with severe SARS-Cov2-induced ARDS. However, well-designed clinical trials are needed to examine this hypothesis in the setting of COVID-19 disease. A phase 2 study of iNO is underway in patients with COVID-19 (NCT04290871) with the goal of preventing disease progression in those with severe ARDS. A phase 3 study (PULSE-CVD19-001) for iNO (INOpulse; Bellerophon Therapeutics) was accepted by the FDA in mid-March 2020 to evaluate the efficacy and safety in patients diagnosed with COVID-19 who require supplemental oxygen before the disease progresses to necessitate mechanical ventilation support.

## 9. The Cytokine Release Storm (CRS)

A defining feature of COVID-19-induced ARDS is the presence of a cytokine release storm (CRS) that is associated with worse outcomes [113,114]. The CRS in COVID-19 is characterized by neutrophilia and lymphocytopenia, as well as elevated levels of interleukin (IL)-1β, IL-2, IL-6, IL-8, IL-9, IL-10, IL-17, Granulocyte colony stimulating factor (G-CSF), Granulocyte-macrophage colony-stimulating factor (GM-CSF), interferon (IFN)γ, tumor-necrosis factor alpha (TNF-A), Interferon gamma-induced protein 10 (IP10_, and Monocyte Chemoattractant Protein-1 (MCP1) [3,113]. Of these cytokines, IL-6 has been identified as one of the most important pro-inflammatory mediators in COVID-19 [115,116]. This has resulted in the use of the IL-6 monoclonal antibody tocilizumab for the treatment of COVID-19 [117,118,119]. In these studies, the use of tocilizumab is recommended as an approach to reduce mortality in COVID-19 patients [117,118]. This is significant to pulmonary vasoconstriction, as IL-6 is well known to promote PH through the activation of Signal transducer and activator of transcription 3 (STAT3) via constitutive or trans-activation pathways [120,121,122,123]. However, it is important to note that its effects are most closely associated with vascular remodeling and less so with acute vasoconstriction. However, IL-6 has been shown to promote vasoconstriction in isolated pulmonary arteries; thus, it is conceivable that in addition to inflammation, it could mediate acute and chronic effects contributing to PH in COVID-19. The A Study to Evaluate the Efficacy and Safety of Remdesivir Plus Tocilizumab Compared With Remdesivir Plus Placebo in Hospitalized Participants With Severe COVID-19 Pneumonia (REMDACTA) study adds tocilizumab to a regimen of remdesivir in hospitalized patients with severe COVID-19 pneumonia. The A Study to Evaluate the Safety and Efficacy of Tocilizumab in Patients With Severe COVID-19 Pneumonia (COVACTA) study is completed enrollment to evaluate tocilizumab plus standard of care versus standard of care alone in patients hospitalized with severe COVID-19. In addition, the EMPACTA (A Study to Evaluate the Efficacy and Safety of Tocilizumab in Hospitalized Participants With COVID-19 Pneumonia) study will focus on trials in sites known to provide critical care to underserved and minority populations.

Another important CRS component that is associated with PH is TNF-A. TNF-A drives PH by suppressing bone morphogenic protein receptor 2 (BMPR2) signaling in pulmonary artery smooth muscle cells [124]. This depletion of BMPR2 is also induced by increased IL-6 signaling [122] and enhanced transforming-induced growth factor (TGF)-β induced vascular remodeling [122,124]. However, similar to IL-6, TNF-A can itself promote vasoconstriction directly [125]. Interestingly, the glycosaminoglycan, hyaluronan, can be upregulated by TNF-A and by IL-1, which is another CRS component [126]. Hyaluronan is elevated in PH [127,128] and the inhibition of hyaluronan by 4-methylumbelliferone has been shown to be effective in PH predominantly by attenuating vascular remodeling [129,130]. However, hyaluronan can also promote cell stiffness through Ras homolog family member A (RhoA) [129]. Thus, an inhibition of hyaluronan using the clinically used inhibitor 4-methylumbelliferone could be a valid strategy for the treatment of COVID-19 [131]. Indeed, increased evidence of hyaluronan upregulation has been reported in RNAseq data from bronchoalveolar cells and peripheral blood mononuclear cells from patients with COVID-19 [132]. The central pathways that promote lung injury following viral infection and how mediators of inflammation induce lung pulmonary vasoconstriction and vascular remodeling are summarized in Figure 2.

## 10. Endothelin

Endothelin is a polypeptide that is produced by the vascular endothelium. Endothelin is a potent vasoconstrictor that induces vascular smooth muscle proliferation and in patients with PAH, high plasma levels of endothelin-1 (ET-1) have been documented due to an increase in production in endothelial cells and decreased elimination of ET-1 in lung [133]. Currently, endothelin receptor antagonists (ERAs) are approved for treatment in pulmonary arterial hypertension by demonstrating efficacy in PAH (Seraphin, ARIES and Breathe trials). The biological action of ET1 is mediated by two G-protein-coupled subtypes of receptors: ETA (endothelin A) and ETB (endothelin B). ETA receptors are expressed on pulmonary smooth-muscle cells, and they mediate potent vasoconstriction and promote cell proliferation. ETB receptors are expressed predominantly on the endothelial surface of vessels, and they mediate vasodilatation through the production of nitric oxide and prostacyclin; they also stimulate pulmonary clearance of circulating ET1. ETB receptors not only have “protective” effects, but they are also present in the muscle cells of vascular walls, where they have the same effects as ETA receptors (vasoconstriction and cell proliferation). In systemic and pulmonary hypertension, the expression of ETB receptors is upregulated in the media of blood vessels, and ETA and ETB receptors contribute to the detrimental effects of ET [134]. 

ERAs not only modulate vascular tone; they can also affect the inflammatory cascade. Bosentan has been shown to significantly reduce pro-inflammatory and profibrotic cytokines such as interleukin (IL) 2, 6, 8 and interferon-γ in scleroderma patients [135]. Bosentan has shown some efficacy against the viruses along with some anti IL-6 properties [57,136]. Bosentan showed a reduction of viral RNA copy number (70–90%) in human umbilical vein endothelial cells. Guo et al. reported a case of a patient who had influenza A-related acute respiratory distress syndrome needing mechanical ventilation. Subsequent to the use of Bosentan, there was marked improvement, leading to the weaning of mechanical ventilation [137]. These observations suggest a role of ERA in the hypoxic vasoconstrictor seen in SARS CoV-2 infection, which warrants further research.

## 11. The Hypoxic-Adenosinergic Response 

The hypoxic-adenosinergic response is one of the key regulators of the adaptive hypoxic responses in ARDS [138]. In this pathway, rapid stabilization of hypoxia inducible factor alpha (HIF-1A) leads to the upregulation of the ecto-nucleotidases CD39 and CD73 [139]. These enzymes convert adenosine triphosphate (ATP) released by erythrocytes following hypoxia [140] or other injury to adenosine [139]. Although the initial surge of adenosine has been shown to be protective in the setting of acute lung injury [141], the prolonged accumulation of adenosine has been shown to be detrimental and contributes to chronic lung injury [142] and PH [143]. Indeed, studies have shown that ATP can promote pulmonary vasoconstriction through the activation of P2Y1 and P2Y12 receptors [144], or in the case of adenosine, through the activation of adenosine A1 receptor [145]. Importantly, despite its protective effects, elevated adenosine through the activation of its A2B receptor (ADORA2B) can promote an increased expression of IL-6 [146,147] and the upregulation of hyaluronan synthase 2 (HAS2), which is a key enzyme that leads to the accumulation of hyaluronan [148]. It is conceivable that in COVID-19, the prolonged and severe hypoxemic response may outpace the protective effects of adenosine. Alternatively, excessive hypoxic conditions may be significant enough in duration to trigger the detrimental effects of adenosine; further data examining the role of adenosine in COVID-19 are necessary to fully establish its role.

## 12. Clotting Cascade and Pulmonary Vascular Microthrombi

Preliminary studies of lung pathology from COVID-19 patients who died from respiratory failure identify both pathological findings of ARDS—such as diffuse alveolar damage, hyaline membrane formation, and interstitial and alveolar edema—and evidence of severe endothelial injury and widespread thrombosis with microangiopathy [31]. The presence of diffuse thrombosis in severe COVID-19 lung disease presents an additional pathophysiological mechanism that may hypothetically contribute to impairments in ventilation–perfusion matching. Microvascular thrombosis is a well-known feature of PH, while dysfunction of the vascular endothelium contributes to the development of pulmonary hypertension in sickle cell disease [149,150]. Direct viral injury of the endothelium in COVID-19 infection may impair the homeostatic regulation of clotting and lead to a propensity to form thrombus. One possible mechanism is through reduced nitric oxide production by an injured endothelium, leading to increased platelet activation and initiation of the clotting cascade [151]. 

The formation of in situ thrombosis may lead to further pulmonary vascular dysregulation. Fibrin monomers and fibrin degradation products induce pulmonary vasoconstriction through the stimulation of thromboxane A2 (TXA2) synthesis [152]. Endothelial damage induced by COVID-19 infection could conceivably lead to a progressive cycle of injury exacerbated by signaling via products of the clotting cascade. Understanding this connection may lead to therapeutic areas to lessen disease severity. Murine models of chronic hypoxia-induced PH demonstrated improved hemodynamics and reduced pulmonary vascular remodeling in animals overexpressing tissue factor pathway inhibitor (TFPI) [153]. A reversible inhibitor of Factor Xa, TFPI, impedes the early stages of the blood coagulation cascade and may be protective against the effects of thrombosis-mediated pulmonary hypertension [154]. In addition, ifetroban, a clinically tested TXA2 inhibitor, has been shown to attenuate vasoconstriction and thrombosis [155], underscoring its potential therapeutic use in COVID-19. Restoring homeostasis to thrombotic pathways may provide a new approach to lessening the severity of severe lung injury in COVID-19 infection. The mechanisms that can promote pulmonary vasoconstriction arising from overactivation of the endothelin system, the hypoxic-adenosinergic response, and thrombotic pathways are detailed in Figure 3.

## 13. Conclusions

We assert the current notion that ARDS in COVID-19 is not the typical ARDS, which has been well studied in literature with its classic stages. Although the unique pathophysiology of COVID-19-induced ARDS is still in its nascent stages, some of its unique features include an imbalance of the RAS, thrombogenic processes, vascular remodeling triggered by endotheliitis, and a cytokine release storm. These unique processes have the capacity of altering the physiology of the vasculature that can lead to PH. COVID-19 deaths are secondary to a dual phenomenon of ARDS, along with severe pulmonary vascular disease with severe endothelial damage and subsequent inability to oxygenate or ventilate the lung. PH has been well described as a poor prognostic indicator in ARDS, but its role in COIVD-19 respiratory failure is not as clear and presents a putative target for therapy or disease amelioration. We have provided a detailed review of the potential mechanisms and therapeutic targets that contribute to PH in COVID-19-induced ARDS focusing on pulmonary vasoconstriction. The pathways include candidates such as hypoxemia-mediated vasoconstriction, distorted angiogenesis, no homeostatic clotting cascade, HAPE, RAS/Kallikrein system, endothelin, NO imbalance, IL-6, and TNF-A. Numerous clinical trials utilizing drugs that have a capacity for impacting pulmonary vasoconstriction are currently underway (Table 1). A list of clinically tested agents that have the potential of being beneficial in COVID-19 by attenuating vascular effects is provided (Table 2).

## Figures and Tables

**Figure 1 ijms-21-08081-f001:**
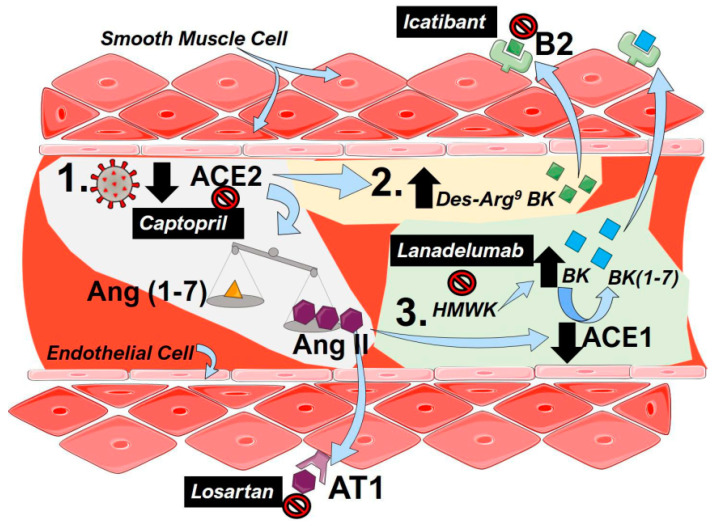
The renin–angiotensin system (RAS) including bradykinin in coronavirus disease 2019 (COVID-19). (1) To infect cells, severe acute respiratory syndrome coronavirus 2 (SARS-CoV-2) binds to angiotensin converting enzyme 2 (ACE2), resulting in a functional loss of enzyme due to internalization and reduced expression of ACE2. This results in an imbalance of the angiotensin (Ang, 1–7, yellow triangles) and Ang (II, purple hexagons), favoring the accumulation of Ang II. Then, Ang II is able to bind to angiotensin receptor 1 (AT1) to mediate pulmonary vasoconstriction and inflammation. (2) Depletion of ACE2 activity following SARS-CoV-2 infection results in an increased a Des-Arg9-Bradykinin (BK, green squares) that can activate the bradykinin (B)2 receptor to promote vasoconstriction. (3) High-molecular-weight-kallikrein (HMWK) is converted to bradykinin (BK, blue squares) that in turn is metabolized to BL (1–7) and other metabolites by ACE1. Increased levels of AngII in COVID-19 reduce the enzymatic capacity of ACE1 resulting in an accumulation of BK that can also activate B2 receptors to promote pulmonary vasoconstriction. Clinically tested agents are shown in black boxes with white font. Captopril can inhibit the effect of ACE2, resulting in lower functional receptors for SARS-Cov-2. Losartan, an angiotensin receptor blocker (ARB), antagonizes the AT1 receptor inhibiting the effects of Ang II. Lanadelumab is a monoclonal antibody against HMWK, preventing its cleavage into BK. Icatibant is a B2 antagonist blocking the effects of both Des-Arg9-BK and BK.

**Figure 2 ijms-21-08081-f002:**
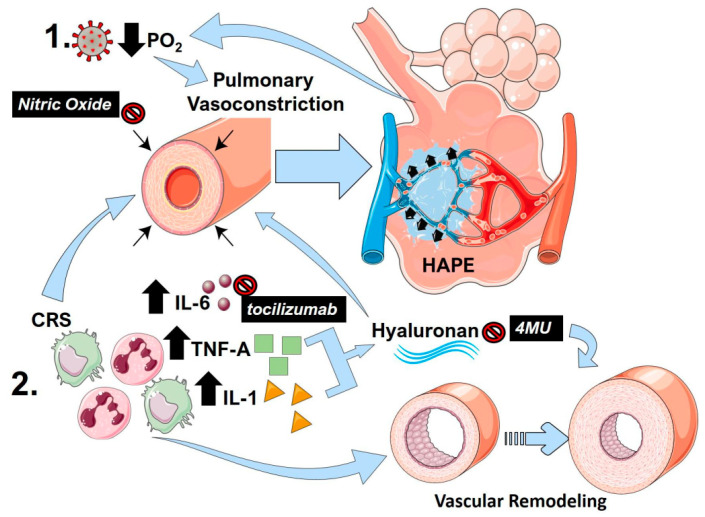
The link between pulmonary vasoconstriction, high altitude-induced pulmonary edema (HAPE) and the cytokine release storm (CRS) in COVID-19. (1) Following SARS-Cov-2 infection, reduced PO_2_ and resulting hypoxia can promote sustained pulmonary vasoconstriction, resulting in pulmonary edema akin to HAPE. This in turn results in reduced PO_2_ that can itself further promote vasoconstriction, resulting in a vicious circle. (2) The cytokine release storm (CRS) in COVID-19 results in an increased in inflammatory cytokines including interleukin (IL)-1 (yellow triangles) IL-6 (brown circles) and tumor-necrosis factor alpha (TNF-A green squares). IL-6 and TNF-A are capable of promoting pulmonary vasoconstriction and vascular remodeling, which are important hallmarks of PH. TNF-A and IL-1 can promote increased hyaluronan levels that also stimulate vascular remodeling and pulmonary vasoconstriction. Clinically tested agents are shown in black boxes with white font. Nitric oxide can promote vasodilation reversing pulmonary vasoconstriction. Tocilizumab is a monoclonal antibody against IL-6, and 4-methylumbelliferone (4MU) inhibits hyaluronan synthesis.

**Figure 3 ijms-21-08081-f003:**
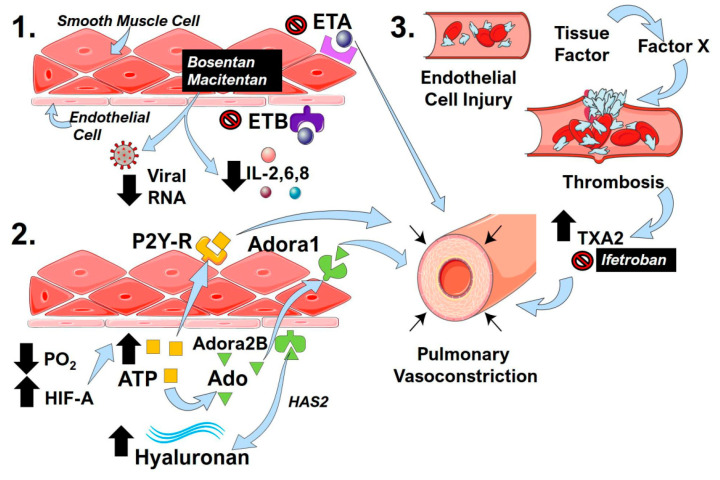
Endothelin receptors, the hypoxic-adenosinergic response, and thrombotic processes in COVID-19 that impact pulmonary contractility. (1) The endothelin receptor antagonists (ERA) bosentan and macitentan inhibit endothelin receptor (ET)A and ETB activity. The activation of ETA is known to induce pulmonary vasoconstriction. ERAs can also reduce inflammatory cytokines such as interleukin (IL)-2,6,8 and cause reduced viral RNA counts. (2) Hypoxemia and resulting low PO_2_ levels in COVID-19 lead to the stabilization of hypoxia inducible factor (HIF)-A, which in turn leads to increased ATP levels (yellow squares). The activation of P2Y-receptors (R) by ATP can promote pulmonary vasoconstriction. ATP can also be converted to adenosine (Ado, green triangles), which can then activate adenosine A1 (Adora1) or Adenosine A2B (Adora2B) receptors. The activation of Adora1 promotes pulmonary vasoconstriction, and the activation of Adora2B results in increased hyaluronan levels through the upregulation of hyaluronan synthase 2 (HAS2). (3) Viral-induced endothelial injury initiates the coagulation cascade in COVID-19 through the upregulation of tissue factor and Factor X that promote thrombosis. The degradation of fibrils during thrombosis leads to increased thromboxane A2 (TXA2), which is a potent vasoconstrictor that can be inhibited by ifetroban.

**Table 1 ijms-21-08081-t001:** Clinical trials on COVID-19 that can impact pulmonary vasoconstriction.

Drug Name	Clinical Trial Registration	Molecular Target	Phase Level	Reference
Nitric Oxide Gas Inhalation	NCT04305457, NCT04290871, NCT0433882, NCT04312243, NCT04306393, NCT03331445, NCT04388683, NCT04383002	Nitric Oxide	Phase 2	[82,90,156,157,158,159,160,161]
RLF-100 (Aviptadil)	NCT04453839	Protects alveolar type II cells from the SARS-CoV-2 virus	Phase 2,3	[162]
Intravenous Aviptadil	NCT04311697	Protects alveolar type II cells from the SARS-CoV-2 virus	Phase 2	[163]
Inhaled Aviptadil	NCT04360096	Protects alveolar type II cells from the SARS-CoV-2 virus	Phase 2,3	[164]
R-107	Waiting for registration	Steadily releases NO into lung tissues	Not Applicable	[165]
Sildenafil	NCT04304313	Inhibitor of cGMP-specific phosphodiesterase (PDE-5)	Phase 3	[85]
Ambrisentan	NCT04393246	Endothelin receptor antagonist, and is selective for the type A endothelin receptor (ETA)	Phase 2	[166]
Losartan	NCT0431117 and NCT04312009	An angiotensin II receptor blocker (ARB)	Phase 2	[167]
Remdesivir	NCT04292899	RNA polymerase inhibitor	Phase 3	[89]

**Table 2 ijms-21-08081-t002:** Clinically tested drugs that affect pulmonary vasoconstriction that could be used in COVID-19.

Drug Name	Target	Other Beneficial Effects	References
4-methylumbelliferone (4MU)	Hyaluronan inhibitor		[129,130]
Captopril	ACE inhibitors		[75,76]
Icatibant	Selective antagonist for the B2 receptor	Reduction of cough symptoms and edema	[63,64]
Ifetroban	Thromboxane A2 antagonist	Inhibition of thrombotic processes	[155]
Lanadelumab	A monoclonal antibody (class IgG1 kappa) against high-molecular-weight kininogen that yields bradykinin	Reduction of cough symptoms and edema	[59,67]
Losartan	An angiotensin II receptor blocker (ARB)	Reduction of inflammation	[77]
Macitentan (ACT-064992), Bosentan	Prevents the binding of ET-1 to both endothelin A (ETA) and endothelin B (ETB) receptors (endothelin receptor antagonist)	Inhibition of inflammation, Reduction of viral RNA counts	[135]

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
