# Peer review of "Emerging Mechanisms of Pulmonary Vasoconstriction in SARS-CoV-2-Induced Acute Respiratory Distress Syndrome (ARDS) and Potential Therapeutic Targets"

_ijms, 2020, doi:10.3390/ijms21218081_

Round 1
Reviewer 1 Report
The article by Karmouty-Quintana et al. deals with very actual problem of SARS-CoV2-induced ARDS. Based on some unique characteristics of COVID19-induced ARDS, authors emphasize participation of imbalance in renin-angiotensin system and kallikrein-kinin system, injury to endothelium, modulation of vascular tone, thrombogenesis, and cytokine release storm. Regarding that ARDS-induced pulmonary hypertension is not just an indicator of poor prognosis, but may represent a potential therapeutic target, authors presented several pathways in relation to pulmonary vasoconstriction which might be therapeutically influenced.
The article is clearly written, and problems of ARDS-induced pulmonary vasoconstriction and pulmonary hypertension are sufficiently discussed. I have no other comments or objections to the authors.
Author Response
Thank you very much for recognizing the impact and significance of our study, we are very pleased for your highly favorable comments.
Reviewer 2 Report
The currevent about the association between COVID-19 and pulmonary hypertension is limited, therefore, I do not think it is too early to perform this review based on too many unclear evidence.
Author Response
Thank you for expressing your concerns. We agree with the fact that COVID-19 is a new disease and as such many of the mechanisms contributing to its patho-physiology are not yet fully understood. The goal of our manuscript is to highlight the potential effects of COVID-19 on the pulmonary vasculature, an area that has not received as much attention.
Reviewer 3 Report
First of all, the manuscript entitled "Emerging mechanisms of pulmonary vasoconstriction in SARS-CoV-2-induced Acute Respiratory Distress Syndrome (ARDS) and potential therapeutic targets" submitted in IJMS by Harry Karmouty-Quintanan et al is well written and discussed. However, I found minor correction before final acceptance. Kindly go through and justify them (File attached)
- Check for editing and spell correction.
- Kindly revised the total infected patient data till date when you submit the revised draft.
- Reference cited need to be improved.
- For correction- check file attached.

Author Response
Thank you very much for your highly favorable critique. We have addressed all four points and we very much appreciate your carefully revised manuscript, incorporating all your changes to the revised manuscript.
Round 2
Reviewer 2 Report
the authors resposne well, so I have no more suggestion.